# Comparing Mental Health, Wellbeing and Flourishing in Undergraduate Students Pre- and during the COVID-19 Pandemic

**DOI:** 10.3390/ijerph19127438

**Published:** 2022-06-17

**Authors:** Marien Alet Graham, Irma Eloff

**Affiliations:** 1Department of Science, Mathematics and Technology Education, University of Pretoria, Pretoria 0002, South Africa; 2Department of Educational Psychology, University of Pretoria, Pretoria 0002, South Africa; irma.eloff@up.ac.za

**Keywords:** student wellbeing, student mental health, flourishing, mental health continuum short form, flourishing scale, fragility of happiness scale, undergraduate students, pandemic

## Abstract

There has been a preponderance of studies on student mental health, wellbeing and flourishing during the COVID-19 pandemic. Few studies have compared data on student mental health and wellbeing before and during the pandemic. The purpose of the current study was to compare mental health and wellbeing in undergraduate students before and during the COVID-19 pandemic. Survey research was conducted with three groups of undergraduate students (n = 905) from diverse scientific fields at a large, urban university in South Africa. Data was collected by means of electronic surveys, combining full-scale items from three instruments, the Mental Health Continuum Short Form, the Flourishing Scale and the Fragility of Happiness Scale. Data was analysed by the Statistical Package for the Social Sciences (SPSS), the Analysis of Moment Structures (AMOS) and R software. The results indicate that while the mental health and wellbeing of students declined during the pandemic concerning their perceived ability to contribute to society, having supportive and rewarding social relationships and them being engaged and interested in their daily activities, it also improved in terms of their perceived ability to manage their daily lives (environmental mastery), being challenged to grow (personal growth) and in terms of their views that society was becoming better (social growth/actualisation).

## 1. Introduction

The magnification of the importance of student wellbeing since the start of the COVID-19 pandemic has been inimitable. The number of studies investigating the wellbeing of a variety of student populations in various regional and national contexts has grown significantly [1,2,3].

Despite wide-ranging concerns about student wellbeing, the findings from empirical studies have been varied. [4] specifically report positive student self-efficacy beliefs within the context of online learning. Similarly, no significant increase in psychological distress and life stress in terms of the online learning environment has been found [5]. However, the continuum of psychological distress as reported by [6] (p. 170) on the “prevalence of no psychological distress (16.67%) followed by mild (40%), moderate (30.56%), and severe psychological distress (12.78%)”, potentially echoes in other contexts as well, with the majority of students presenting mild to moderate psychological distress and smaller groups of students experiencing either severe distress or no distress at all.

It should be noted that numerous studies [7,8,9] have distinguished between the wellbeing of various student populations within the broad undergraduate student population. These studies have indicated dissimilarities in student populations relating to gender differences, field of study and year of education. The complexities of the wellbeing of students in the health sciences have, for instance, been especially pertinent. In a study that mapped the complex experiences of dental students in years three to five [8], women were found to be significantly more stressed due to a lack of clinical skills (*p* = 0.048). The study also highlighted elevated stress levels due to a lack of existing clinical skills (*p* = 0.012) and challenges in terms of being able to complete the required clinical skills courses. Students in this study expressed concerns about “not being a good enough dentist” after graduation (*p* = 0.002) [8] (p. 1). This study indicates that the highest stress levels were experienced by fourth year and female dental students. Gender differences in undergraduate student populations have also been indicated, with male nursing students scoring higher on eHealth literacy [9] and female students, in another study, reporting “a decrease in feeling tired or having little energy (*p* = .03), and a decrease in having poor appetite or overeating (*p* = .04) [7] (p. S77). Females further reported an increase in feeling useful (*p* = .04) and dealing with problems well (*p* = .04)” [7] (p. S77).

Despite the preponderance of studies on student wellbeing involving various student populations during the pandemic, few studies have been undertaken that compare pre-pandemic levels of student wellbeing with student wellbeing during the pandemic. Predominantly, student wellbeing studies have collected and analysed data during the pandemic, which limits comparisons with pre-pandemic indicators of student wellbeing. Despite the fact that studies indicate fluctuating levels of student wellbeing during the pandemic as well as the need for more nuanced discernment between the wellbeing of different undergraduate student populations, there is a need to understand the directionality of the shifts in student wellbeing compared to pre-pandemic levels. It should also be noted that within the recent increase in outputs in this field, the quality of studies fluctuates considerably, meaning that replicability may be challenging, and sample sizes also vary significantly.

### 1.1. Rationale for the Study

This study sought to compare the wellbeing of a diverse population of undergraduate students before the COVID-19 pandemic with the wellbeing of (a comparable population of) undergraduate students during the COVID-19 pandemic. The rationale for the study was to contribute to the understanding of student wellbeing in order to optimise student support. In understanding the specific domains within which student wellbeing has increased or decreased, more tailored interventions to support students may potentially be developed.

### 1.2. Hypotheses

Ho: There are no statistically significant differences in terms of mental health and wellbeing between three groups of undergraduate students, pre- and during the COVID-19 pandemic.

Ha: There are statistically significant differences in terms of mental health and wellbeing between three groups of undergraduate students, pre- and during the COVID-19 pandemic.

### 1.3. Ethics Approval

The study received ethical clearance (GW0180232HS) from the Ethics Committee of the Faculty of Humanities at the university where the study was conducted.

## 2. Method

With the objective of this study being to compare the mental health and wellbeing of undergraduate students before and during the COVID-19 pandemic using three groups of students, a large urban residential university in South Africa was used as the site of study. [10] conducted a comparative analysis at this university pre-COVID using two groups of students; however, for the purpose of the current study, additional data was collected from a third group of students during COVID. This comparative dataset was used in the current study to compare the mental health and wellbeing of undergraduate students before and during the COVID-19 pandemic. The university offers approximately 1,200 academic programmes to undergraduate and postgraduate students in the social sciences, natural sciences and a variety of professional degree programmes, such as health sciences, engineering, law and education. The student body is highly diverse and includes national and international students, and the language of instruction is English. The academic year is divided into two semesters: from January to June and from July to December.

### 2.1. Study Setting and Data Collection

For the first survey (S1) and the second survey (S2), the instruments were distributed electronically to undergraduate students who were formally registered at the university and residing in university accommodation. Student leaders, responsible for student wellbeing in their portfolios, invited students to participate. In total, 551 undergraduate students responded to the first survey (data collected in February 2019) and 281 to the second survey (data collected from September to October 2019). For the third survey (S3), which was conducted during the pandemic (data collected from September to December 2021), the questionnaires were again distributed via student leadership structures, with supplementary distribution through the offices of deputy deans. In the third survey, a total of 293 students from the same university responded. The lower response rate for the latter two surveys was attributed to the fact that they were conducted at the end of the year, close to and during the examination period. The students’ student numbers were captured, purely to see whether the groups were related or unrelated, and since only four of the students in the third study had also completed one or both of the earlier surveys, it was decided to only work with the unrelated groups, which led to the sample sizes of the studies being 443, 173 and 289, respectively, resulting in a total sample size of 905. In this way, comparative analysis of the same three instruments for unrelated groups of undergraduate students, before and during the COVID-19 pandemic, was possible. The current study thus expands on an ongoing study [10] that measures undergraduate student wellbeing at different time points.

### 2.2. Design

Three instruments, namely, the Mental Health Continuum Short Form (MHC-SF), the Flourishing Scale (FS) and the Fragility of Happiness Scale (FOHS), were combined as an electronic survey and distributed via email using Qualtrics. The MHC-SF consists of 14 items and has been used in South Africa before [11]. Of the 14 items, three assess emotional wellbeing (MHC-SF-E), five social wellbeing (MHC-SF-S) and six psychological wellbeing (MHC-SF-P). The MHC-SF measures on a 6-point Likert scale (1 = “never”; 2 = “once or twice”; 3 = “about once a week”; 4 = “about two or three times a week” 5 = “almost every day”; and 6 = “every day”) with higher individual values indicating better mental health. The overall score (the MHC-SF overall score ranges from 14 (all responses from one student is “never”) to 84 (all responses from one student is “every day”), and, using a similar argument, the overall scores for MHC-SF-E, MHC-SF-S and MHC-SF-P range from 3 to 18, 5 to 30 and 6 to 36, respectively). The FS assesses psychosocial flourishing and is designed to measure social-psychological prosperity [12]. It provides a single psychological wellbeing score (between 8 and 56). Consisting of eight items, psychological needs such as the need for competence, optimism, self-reliance, self-esteem and purpose are measured by the FS on a 7-point Likert scale (ranging from “strongly disagree” to “strongly agree”), with higher individual values indicating higher social-psychological prosperity. The FS overall score ranges from 8 (all responses from one student is “strongly disagree”) to 56 (all responses from one student is “strongly agree”). The FOHS consists of four items using a 7-point Likert scale which is similar to the FS and measures the fragility of happiness beliefs ([13]), with higher individual values indicating higher belief that happiness is fragile. The FOHS overall score ranges from 4 (all responses from one student is “strongly disagree”) to 28 (all responses from one student is “strongly agree”). For all three of these instruments, the reliability and validity have been established in various settings [13,14,15,16,17,18].

### 2.3. Data Analysis

All data analyses were performed using the Statistical Package for the Social Sciences (SPSS), version 28, except for the confirmatory factor analysis (CFA), which was conducted using Analysis of Moment Structures (AMOS), version 28, and for the multivariate Kruskal–Wallis (KW) tests, which were conducted using R software, version 4.2.0. The factor/construct structures of the instruments were assessed using CFA. The reliability was established using the Cronbach’s alpha coefficient. An acceptable level for the Cronbach’s alpha coefficient is 0.7 [19], which indicates that a questionnaire has internal consistency. The validity was established using construct validity consisting of convergent and discriminant validity, respectively. Convergent validity shows that items that load onto the same construct correlated significantly with each other. Discriminant validity, on the other hand, shows that items that do not load onto the same construct correlate with each other less strongly than items belonging to the same construct. Robust, nonparametric Spearman correlations were used and, for conciseness, not all correlations are presented in this article; however, it was found that items loading onto the same construct correlated more strongly with each other than those belonging to different constructs. For the CFA, in the past, to access goodness-of-fit (GOF), the Chi-square statistic and the Goodness-of-Fit Index (GFI) were used. However, the Chi-square statistic is very sensitive to the sample size and is no longer relied upon as a basis for acceptance or rejection [20,21,22], and, given the sensitivity of the GFI, it has become less popular [20], and it has even been recommended that it not be used [23]. Some remaining GOF measures are the root mean square error of approximations (RMSEA), the comparative fit index (CFI) and the Tucker–Lewis index (TLI), and they are considered here. For the RMSEA, recommendations vary, with [24] stating that values above 0.10 indicate a poor fit, values between 0.08 and 0.10 provide a mediocre fit and values below 0.08 show a good fit, [25] stating that the values should be above 0.07 and [26] stating that the values should be above 0.06. Summarising all these different recommendations, [20] concluded that, for a well-fitting model, the RMSEA will be close to zero and it should not exceed 0.08; this is similar to the recommendation of [27]. For the CFI, certain studies recommend a CFI > 0.95 [26,28]. Others recommend a TLI > 0.90 [29] and a TLI > 0.95 [26].

Since the overall scores of the constructs are continuous variables, the Kolmogorov–Smirnoff test was used to test the normality, and the data was found to differ significantly from normality (*p* < 0.05); accordingly, nonparametric multivariate Kruskal–Wallis (KW) tests and nonparametric Spearman correlations were used. Since the data differed significantly from normality, instead of only reporting on the mean and standard deviation, which is typically for normally distributed variables, in the current study, two measures of location (the mean ($\bar{x}$) and median ($\tilde{x}$)) and two measures of spread (the standard deviation (SD) and the interquartile range (IQR)) were considered to provide a more thorough picture of the underlying distributions. Ideally, if the data were normally distributed, the multivariate analysis of variance (MANOVA), which assesses multiple dependent variables simultaneously for three or more groups, would have been used; however, due to the data being non-normal, the nonparametric multivariate KW tests [30] with ad-hoc tests were used. For ad-hoc tests and the multivariate KW test and for the individual Likert scale items and overall scores, the KW statistic was used to test for differences between the three independent groups (S1, S2 and S3), and Spearman correlations were used to compute correlations. When the *p*-value of the KW test indicates significant differences (*p* < 0.05), Dunn’s test is used for post-hoc pairwise testing. It should be noted at this point that some studies in the literature state that one can conduct pairwise Mann–Whitney (MW) tests as a post-hoc for the KW test. Running pairwise MW tests following the KW test is incorrect, as the ranks used for the pairwise MW tests are not the ranks used for the KW test and MW tests do not use the pooled variance implied by the KW null hypothesis [31]; Dunn’s test does not have these problems [32]. The reason why this is highlighted is that the results of the pairwise Dunn’s test should not be compared with the results of the pairwise MW tests performed. For example, in the article by [10], for the item (mhc9) “That you liked most parts of your personality”, using the MW test, significant differences were found when only comparing two independent groups (S1 and S2, called baseline and follow-up unrelated groups in [10]. However, here (see Section 3.3), when comparing three independent groups, no significant differences were found. Thus, although the same data was used in both studies for S1 and S2, different statistical tests were performed, as the current article considered a third timepoint (S3). It should also be noted that some studies make use of the Bonferroni adjustment for multiple comparisons when conducting post-hoc tests; however, in the current study, this was not performed, as the use of these types of adjustments have been criticised in the literature; see, for example, [33]. For the nominal categorical biographical variables, the two-proportions z-test was used to establish whether differences between column proportions (percentages) were statistically significant (*p* < 0.05) or not (*p* > 0.05). In an earlier analysis on the pre-pandemic dataset [10], the Chi-square statistical technique was used for the comparison of the nominal categorical biographical variables, since only two groups were compared. In the current study, however, pairwise z-test comparisons were used to ascertain statistically significant differences between the three groups (between S1 and S2, S1 and S3 or between S2 and S3).

## 3. Results

### 3.1. Biographical Variables of Participants

A summary of the biographical variables is provided in Table 1, and only the cases where there are significant differences are discussed here. It is interesting to note that the proportion of missing values for S3 is, for each biographical variable, significantly higher than the proportion of missing values of S1 and S2, respectively. Out of the 289 respondents for S3 that did complete the questions for the MHC-SF, FS and FOHS, 20 of them (6.9%) did not complete the biographical section. For the S3 questionnaire, in order to increase inclusivity, the gender variable the options “prefer not to disclose” and “prefer to self-describe” were added, which were not options in the earlier two questionnaires. Yet, 6.9% of S3 respondents did not feel comfortable completing the biographical section of the questionnaire.

There is only one province for which a significant difference was found, namely, the Eastern Cape, where the percentage for S3 (6.2%) is statistically significantly higher than that of S1 (2.5%) and S2 (1.2%), respectively (S1: z = 2.366; *p* = 0.018; S2: z = 2.346; *p* = 0.019). In the case of gender, S1 had significantly more males (35.9%) when compared to S2 (24.3%) and S3 (14.5%), respectively (S2: z = 2.697; *p* = 0.007; S3: z = 6.244; *p* < 0.001), which then also means that S1 had significantly fewer females compared to the other groups. Regarding race, S3 had significantly more African students (46.7%) when compared to S1 (35.9%), but the difference was not significant with S2 (39.3%) (S1: z = 3.291; *p* = 0.001; S2: z = 1.542; *p* = 0.123). S3 also had significantly fewer white students (38.4%) compared to S1 (54.2%) and S2 (50.9%), respectively (S1: z = 2.968; *p* = 0.003; S2: z = 2.512; *p* = 0.012). This trend shows a significant decline in white students and an increase in the response rate of African students over time. In terms of the home language, S3 had significantly fewer (22.5%) Afrikaans-speaking students when compared to S2 (35.3%) but not significantly fewer than S1 (30.5%). This shows a significant decline in the number of Afrikaans-speaking students from September–October 2019 (S2) to September–December 2021 (S3). In the case of Venda-speaking students, the percentages for S1, S2 and S3 were 0.9%, 3.5% and 2.1%, respectively, where the difference between S1 and S2 is statistically significant (z = 2.226; *p* = 0.026). Except for the number of missing values (discussed earlier), there were no significant differences between the three groups when it comes to citizenship.

### 3.2. Reliability and Validity of the Instruments

To test the reliability of the three instruments, Cronbach’s alpha coefficient was used, and the results are provided in Table 2. As stated, the MHC-SF is comprised of three separate constructs, namely, emotional, psychological and social wellbeing, and, accordingly, the Cronbach’s alpha coefficients for each construct as well as for the entire MHC-SF scale are provided.

From Table 2, it is clear that all three instruments are reliable. Next, a CFA was conducted to establish the factor/construct structures of the three instruments; the results are provided in Table 3. Once established, the validity was established using construct validity (see Section 2.3). For the MHC-SF, it should be noted that the CFA for the 3-factor MHC-SF was run, and the GOF measures were below the acceptable standards. However, studies have repeatedly shown that the bifactor model for the MHC-SF outperforms the 3-factor model [35,36], and, accordingly, Table 3 reports on GOF measures of the bifactor model; for more details on the bifactor model, see the studies conducted by [35] and [36]. For the FS, it should be noted that the model was modified by correlating two pairs of error terms in a step-wise manner: first, e2–e7 (e2: “My social relationships are supportive and rewarding”; e7: “I am optimistic about my future”), followed by e6–e8 (e6: “I am a good person and live a good life”, e8: “People respect me”). This modification was made because the GOF measures were below the acceptable standards without it. This type of modification has been made in practice for the FS; see, for example, the articles by [37] and [38], in which they modified the FS model by correlating one error term and five error terms, respectively. This modification involves examining the modification indices, and the single-factor model showed a better fit when freeing the error covariance of item e2 “My social relationships are supportive and rewarding” and item e7 “I am optimistic about my future” and that of item e6 “I am a good person and live a good life” and item e8 “People respect me”. All the GOF measures were within the acceptable range, except in the case of the RMSEA of the FOHS, which was high. In this regard, [39] have pointed out that the RMSEA has serious problems with simpler models with few degrees of freedom (*df*) and should not be computed for small df models; the FOHS has the smallest *df* of the three models (*df* = 2) in the current study. The CFI and the TLI of the FOHS were within an acceptable range.

After establishing acceptable GOF measures for the basic unconstrained models, the measurement invariance (MI) across the three groups was investigated using multi-group CFA. By establishing the MI, it was indicated that the same underlying construct was being measured across groups, i.e., the models hold across groups (S1, S2 and S3, respectively). Establishing the MI addresses the question of whether the different groups of respondents interpreted a given instrument (measuring a construct) in a conceptually similar manner. There are essentially four hierarchical levels of MI (configural, metric, scale and residual (strict) invariance), and each of the levels builds upon the previous one by introducing additional constraints. The details of each level of invariance are omitted here for conciseness, and the interested reader is advised to refer to [40]; however, it is pointed out that, in the current study, the strictest level, residual invariance, was not considered, as it is not a prerequisite for testing differences between groups [40], with this being the objective of the current study. The fit of different MI models is typically evaluated by comparing the fit of two nested models, and the difference in fit is attributed to the imposed constraints. Adding constraints to each model (as we move up in the hierarchical structure) decreases the fit, and one would ideally want to see that the decrease in model fit is not statistically significant. [40] listed a number of different recommendations for an acceptable decrease in fit ranging from −0.02 in the case of ΔCFI and +0.03 in the case of ΔRMSEA in terms of metric invariance and −0.01 in the case of ΔCFI and +0.015 in the case of ΔRMSEA in terms of scalar invariance, and, without providing all the details for all the models with respect to the MHC-SF, the FS and the FOHS, respectively, MI was established for the instruments in the current study, indicating that the different groups of respondents interpreted a given instrument (measuring a construct) in a conceptually similar manner.

### 3.3. Findings on Student Mental Health and Wellbeing

For the multivariate KW tests, the results were *χ*^2^ = 60.940 and *p* < 0.001, indicating statistically significant differences between the groups. For the individual items and the overall score, the results of the KW test with Dunn’s post-hoc pairwise comparisons (if applicable) for the MHC-SF are summarised in Table 4, and the means and standard errors for the individual items are presented in Figure 1. All direct quotes in Table 4 and Figure 1 are from [41] (pp. 12–13).

Figure 1 and Table 4 indicate that there were statistically significant differences for 11 of the 14 items between the responses of the first group and the follow-up groups.

In the context of Figure 1, the 11 items where significant differences were found will be discussed. For 5 of the 11 items, there is a clear decreasing pattern over time. These items are “Happy”, “Satisfied”, “That you had something important to contribute to society”, “That you belonged to a community” and “That your life has a sense of direction or meaning to it”. The different groups of students, therefore, had significantly lower scores for these five items over time. However, for the other remaining six items, there was an initial decrease from S1 to S2, followed by an increase from S2 to S3, with these increases being significant for three of the items, namely, “That our society is becoming a better place”, “Good at managing the responsibilities of your daily life” and “That you had experiences that challenged you to grow or become a better person”.

This may indicate an upward trend between two similar time points in the academic year, pre- and during the pandemic on specific components of mental health as measured by the MHC-SF. For the MHC-SF overall score, the descriptive statistics for S1 were $\bar{x}$ = 59.38, $\tilde{x}$ = 62.00, SD = 13.72, IQR = 17.00, for S2 were $\bar{x}$ = 53.97, $\tilde{x}$ = 56.00, SD = 14.15, IQR = 19.00 and for S3 were $\bar{x}$ = 54.88, $\tilde{x}$ = 57.00, SD = 14.34, IQR = 21.00, with the pairwise Dunn’s test showing significant differences between S1 and S2 and between S1 and S3, respectively. For the MHC-SF-E, MHC-SF-S and MHC-SF-P, without providing all descriptive statistics for conciseness, this pattern continues. Thus, from S1 to S2 and from S1 to S3, the decreases are significant, indicating that the overall mental health of students decreased significantly from the start of the study compared to later timepoints in the study. It should, however, be noted at this point that the differences between S1 and S2 constitute general effects of the first year of university, specifically, differences in the outcomes between students in their first semester and students in their second semester, studied in-depth by [10]. Both the S1 and S2 assessments were conducted in 2019 before COVID (i.e., these results are free from the pandemic-effect). The students assessed at S3 were all assessed in their second semester (as were the students from S2); to ensure fair comparisons to investigate the pandemic-effect, a focused discussion on S2 vs. S3 is of great interest. In that regard, for the MHC-SF, significant differences between S2 and S3 can be found in 4 of the 14 items, with only one showing a decrease (that students perceived they had something to contribute to society) and three showing an increase (that students perceive that society is becoming a better place, that they perceive themselves good at managing responsibilities of daily life and that they had experiences that challenged then to grow and become better people). Although one item showed a significant decline in mental health due to the pandemic-effect, three items showed significant improvement in terms of mental health despite the pandemic.

Next, for the individual items and the overall score, the results of the KW test with Dunn’s post-hoc pairwise comparisons (if applicable) for the FS are presented in Table 5 and the means and standard errors for the individual items are presented in Figure 2. All direct quotes in Table 5 and Figure 2 are from [42] (p. 1).

From Figure 2 and Table 5, statistically significant differences were found for six of the eight items on the FS and for the overall score.

In the context of Figure 2, the six items where significant differences were found will be discussed. For all these six items (“I lead a purposeful and meaningful life”, “My social relationships are supportive and rewarding”, “I am engaged and interested in my daily activities”, “I am competent and capable in the activities that are important to me”, “I am a good person and live a good life” and “I am optimistic about my future”), the decreases from either S1 or S2 (or both) to S3 are significant, showing a significant decrease over time. Again, noting that the focus should be on comparing S2 and S3 when investigating the pandemic-effect, two items (“My social relationships are supportive and rewarding” and “I am engaged and interested in my daily activities”) had significantly lower social-psychological prosperity during COVID. For the FS overall score, the descriptive statistics for S1 were $\bar{x}$ = 47.51, $\tilde{x}$ = 48.00, SD = 5.87, IQR = 6.00, for S2 were $\bar{x}$ = 46.11, $\tilde{x}$ = 48.00, SD = 7.38, IQR = 8.00 and for S3 were $\bar{x}$ = 44.00, $\tilde{x}$ = 46.00, SD = 8.09, IQR = 10.00, with the pairwise Dunn’s test showing significant differences between all the time points. Thus, the significant decrease from the time period before COVID to the time period during COVID shows that the overall social-psychological prosperity (e.g., flourishing) of students decreased during COVID.

In summary, the individual items, the overall score and the results of the KW test with Dunn’s post-hoc pairwise comparisons (if applicable) for the FOHS are listed in Table 6, and the means and standard errors for the individual items are presented in Figure 3. All direct quotes in Table 6 and Figure 3 are from [13] (p. 1192).

No significant differences were found in the responses between any of the individual items for the FOHS. In terms of the overall scores, no significant differences were found on the FOHS either (Figure 3 and Table 6); for the latter, the descriptive statistics for S1 were $\bar{x}$ = 19.88, $\tilde{x}$ = 20.00, SD = 5.22, IQR = 7.00, for S2 were $\bar{x}$ = 19.84, $\tilde{x}$ = 21.00, SD = 5.52, IQR = 8.00 and for S3 were $\bar{x}$ = 20.27, $\tilde{x}$ = 21.00, SD = 5.53, IQR = 9.00. The FOHS was the only scale in the current study where the responses remained very similar between the three timepoints. The FOHS measures beliefs about the fragility of happiness. Thus, the belief that happiness is fleeting and can easily be supplanted by other emotive states remained consistent for respondents in all three of the groups, i.e., before and during the pandemic. This may mean that existing beliefs about the fragility of happiness were merely confirmed during the pandemic. However, it is perhaps meaningful that no significant increases in beliefs about the fragility of happiness were indicated during the pandemic.

The correlations between the overall scores are shown in Table 7. It indicates that there are statistically significant positive correlations between the MHC-SF and FS overall scores. This is to be expected since the higher overall score for MHC-SF indicates higher levels of student mental health and wellbeing. In the same way, the higher overall score for the FS is indicative of higher social-psychological prosperity (e.g., flourishing). In general, flourishing is regarded as optimal in terms of mental health and wellbeing. It can be assumed from this correlation that a person with good mental health will concurrently also experience social-psychological prosperity (e.g., flourishing) as measured by the FS. As expected, in turn, the overall score on the FOHS indicates a statistically significantly negative correlation with the other two scores.

## 4. Discussion

The results indicate that while the mental health and wellbeing of students declined during the pandemic concerning their perceived ability to contribute to society, having supportive and rewarding social relationships and them being engaged and interested in their daily activities, it also improved in terms of their perceived ability to manage their daily lives (environmental mastery), being challenged to grow (personal growth) and in terms of their views that society was becoming better (social growth/actualisation). For the purpose of this discussion, the analysis will focus primarily on S2–S3 in the current study. It will also focus on the differences in the results on mental health (MHC-SF) and flourishing (FS) specifically. As stated, the pairwise comparisons between S1 and S2 have been reported previously [10] and have highlighted the significant decline in student mental health over the course of the academic year (pre-pandemic). The current analysis seeks to elucidate the pre-pandemic and “within pandemic” results.

Results from the Mental Health Continuum Short Form (MHC-SF) present a fluctuating pattern of mental health for the participants in this study. Within the S2–S3 pairwise comparison, significant differences occur on four items. For one of the items, a significant decline is detected between S2 (pre-pandemic, second semester) and S3 (during pandemic, second semester). In this comparison, students within the pandemic were significantly less inclined to feel that they had something important to contribute to society. For three of the items, a significant incline is, however, also detected where students were significantly more of the view that society is becoming a better place for all people (social growth/actualisation), that they were good at managing the responsibilities of their daily lives (environmental mastery) and that they had experiences that challenged them to grow or become better people (personal growth). The triad (and direction) of these four significant responses may depict despair with the state of the world, a sense of being overwhelmed with the demands of everyday life during the pandemic, and yet, at the same time, may also depict a positive future outlook in terms of their personal contributions, and, perhaps, even agency and deepened meaning-of-life experience. This finding supports previous findings [43] that meaningfulness may potentially predict positive wellbeing but that it is not necessarily predictive of negative wellbeing and that a crisis of meaning is a strong predictor for both positive and negative wellbeing.

Significant differences occurred for two items between S2 (pre-pandemic, second semester) and S3 (during pandemic, second semester), i.e., the S2–S3 pairwise comparison. When students answered the question “My social relationships are supportive and rewarding” and “I am engaged and interested in my daily activities”, the results indicate significant negative differences from S2 (pre-pandemic, second semester) to S3 (during pandemic, second semester), thereby indicating that for students in the second semester during the pandemic significant declines in mental health and wellbeing could be detected in comparison to the mental health of students prior to the pandemic in the second semester.

Results on the Flourishing scale (FS) overall score from S2 to S3 confirm a significant and consistent downward trend, although we note here that, when investigating the FS items separately, only two of the eight items showed a significant decrease from S2 to S3. For this instrument, there was a downward trend for all items, thereby suggesting a comprehensive downward trend in student flourishing. Since flourishing is often regarded as indicative of high levels of wellbeing, these results appear to present an interesting conceptualisation of student wellbeing during a pandemic. Whereas the results from the MHC-SF indicate declines in student mental health and wellbeing concerning students’ perceived contribution to society, they still contain indications of instances where student wellbeing may have increased in terms of certain specific aspects (managing their daily lives, being challenged to grow, and that society is viewed as becoming better) during the pandemic. However, in terms of flourishing (as measured by the FS), the declines are consistent across all items, thereby indicating a waning in student flourishing. Flourishing has been studied extensively in undergraduate student populations prior to the pandemic [44,45,46,47]. In studies on undergraduate student flourishing and languishing [48], it has specifically been indicated that when students have to deal with practical challenges, are overwhelmed by academic demands and experience social isolation, languishing entails. In turn, a secure social support system, independence and academic achievement have been associated with flourishing [48]. The results from the current study appear to support similar indications in light of contextual factors that arose as a result of the pandemic.

The integration of wellbeing into core curricula and the destigmatisation of mental health practices have been suggested for undergraduate student populations during the pandemic [49]. The current study suggests that the presence of “levers” for wellbeing (managing their daily lives, being challenged to grow and that society is viewed as becoming better) may assist in improving more specific aspects of the broad domain of student mental health and wellbeing. The study also suggests that the re-establishment of environments that can facilitate *flourishing* in undergraduate students also needs to be prioritised. This might constitute environments that provide opportunities for social engagement, regulating academic demands and providing support in the context of the practical challenges of tertiary education.

## 5. Limitations of the Study

The fact that the current study considered unrelated groups may be viewed as a limitation, as casual relationships are typically established using longitudinal studies. However, at the same time, the fact that the study investigated unrelated groups also buffers the effects that maturation and instrument familiarity may have had on the responses. The study was also limited by the constraints of data collection during a pandemic. Despite numerous reminders sent to students, and additional distribution through the offices of deputy-deans, the third group remained fairly small and also contained missing values, which reduced the dataset further. However, this limitation was balanced by the possibilities for comparative analysis with a pre-pandemic dataset on the same instruments. A further limitation of the study is the lack of supplementary qualitative data to enrich an understanding of the significant differences that did emerge and the reasons why certain aspects of wellbeing appeared to have improved. Further studies are recommended in this regard. Other limitations include not being able to control the heightened awareness of the importance of student wellbeing during a pandemic and the ways in which it may have impacted student responses and the effect of student fatigue, also as a result of the pandemic.

Future research might include qualitative studies that explore the aspects of student wellbeing that improve, in comparison to declining dimensions of wellbeing. Studies that utilise interviews, focus groups and vignette research may be especially beneficial.

## 6. Conclusions

The current study advocates a granular understanding of the mental health and wellbeing of undergraduate students at university before and during the pandemic. The indications are that student mental health and wellbeing may have decreased during the pandemic in certain ways but that it also increased in terms of specific aspects of mental health and wellbeing, e.g., managing their daily lives, being challenged to grow and that society is viewed as becoming better. The study also indicates that while variation occurs in terms of decreases and increases in mental health and wellbeing, student flourishing declined during the pandemic for students in this study.

## Figures and Tables

**Figure 1 ijerph-19-07438-f001:**
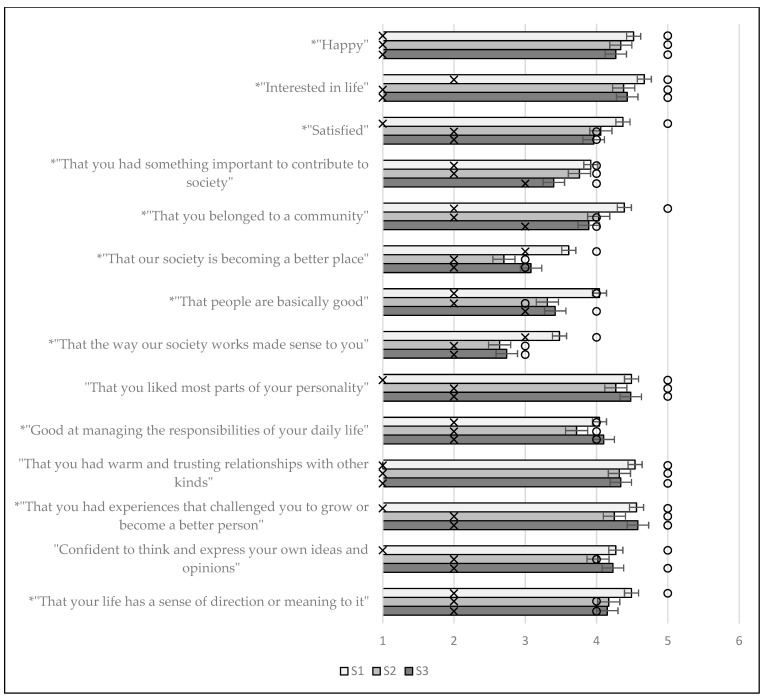
Means and standard error bars for the individual items of the MHC-SF; the IQR is represented by an ‘x’, the median by a circle and an asterisk indicates a statistically significant difference in one or more groups.

**Figure 2 ijerph-19-07438-f002:**
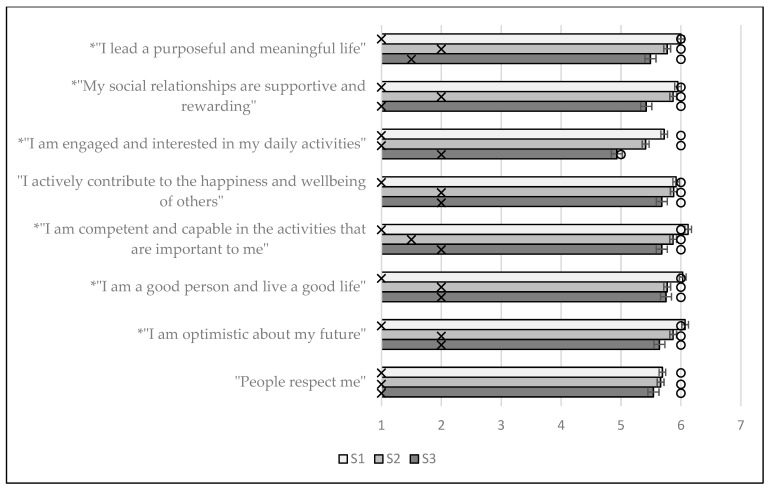
Means and standard error bars for the individual items of the FS; the IQR is represented by an ‘x’, the median by a circle and an asterisk indicates a statistically significant difference in one or more groups.

**Figure 3 ijerph-19-07438-f003:**
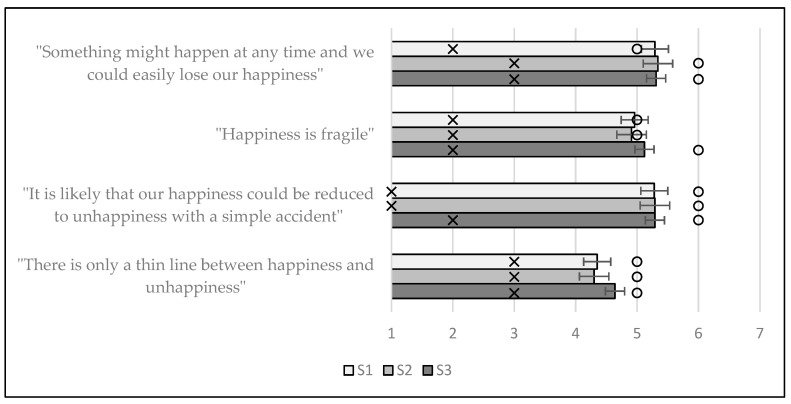
Means and standard error bars for the individual items of the FOHS; the IQR is represented by an ‘x’ and the median by a circle.

**Table 1 ijerph-19-07438-t001:** Cross-tabulation of biographical variables (% within group).

Province	S1	S2	S3
Eastern Cape *	2.5%	1.2%	6.2%
Free State	2.7%	4.0%	4.8%
Gauteng	50.6%	48.0%	37.4%
KwaZulu-Natal	15.1%	11.6%	14.9%
Limpopo	10.4%	11.6%	10.7%
Mpumalanga	10.6%	12.1%	11.1%
North West	5.9%	6.9%	3.8%
Western Cape	2.3%	4.6%	2.8%
Northern Cape	0.0%	0.0%	1.4%
Missing *	0.0%	0.0%	6.9%
**Citizenship**	**S1**	**S2**	**S3**
SA Citizen	94.6%	97.1%	90.0%
SADC Country	2.7%	2.3%	2.4%
Non-African Student	1.8%	0.6%	0.3%
Other African Country	0.9%	0.0%	0.3%
Missing *	0.0%	0.0%	6.9%
**Gender**	**S1**	**S2**	**S3**
Male *	35.9%	24.3%	14.5%
Female *	63.7%	74.6%	75.1%
Prefer not to disclose	Was not an option in this questionnaire		2.8%
Prefer to self-describe	Was not an option in this questionnaire		0.7%
Missing *	0.5%	1.2%	6.9%
**Race**	**S1**	**S2**	**S3**
African *	35.9%	39.3%	46.7%
Coloured **	4.1%	4.0%	2.8%
Indian	3.6%	4.6%	3.1%
White *	54.2%	50.9%	38.4%
Other	2.3%	1.2%	2.1%
Missing *	0.0%	0.0%	6.9%
**Home Language**	**S1**	**S2**	**S3**
Afrikaans *	30.5%	35.3%	22.5%
English	34.8%	29.5%	24.6%
IsiZulu (Zulu)	5.9%	9.8%	10.0%
Northern Sotho (Sepedi)	7.2%	9.2%	9.7%
Setswana (Tswana)	5.6%	3.5%	6.2%
Tshivenda (Venda) *	0.9%	3.5%	2.1%
Sesotho (Southern Sotho)	3.4%	1.2%	3.8%
SiSwati (Swati)	2.0%	0.6%	3.1%
IsiNdebele (Ndebele)	0.7%	0.6%	1.0%
Xitsonga (Tsonga)	0.9%	2.9%	2.8%
IsiXhosa (Xhosa)	3.8%	1.7%	3.8%
Other	4.3%	2.3%	3.5%
Missing *	0.0%	0.0%	6.9%

* An asterisk indicates a statistically significant difference in one or more groups; ** The study recognises the contested nature of the term “coloured” [34] and resists essentialist uses of the term.

**Table 2 ijerph-19-07438-t002:** Cronbach’s alpha coefficients for the three instruments.

Instrument	Cronbach’s Alpha
MHC-SF—entire	0.925
MHC-SF—emotional	0.880
MHC-SF—psychological	0.832
MHC-SF—social	0.856
FS	0.872
FOHS	0.830

**Table 3 ijerph-19-07438-t003:** GOF measures of the CFA for the three instruments.

Instrument	Model *df*	RMSEA	CFI	TLI
MHC-SF	60	0.064	0.969	0.952
FS	18	0.079	0.964	0.950
FOHS	2	0.126	0.979	0.936

**Table 4 ijerph-19-07438-t004:** Results of the MHC-SF.

Item	KW; *p*-Value	Pairwise Comparisons Dunn’s Test; *p*-Value		
During the Past Month, How Often Did You Feel:	S1–S2–S3	S1–S2	S2–S3	S1–S3
“Happy”	13.503; 0.001 *	−2.409; 0.016 *	−0.473; 0.636	−3.411; <0.001 *
“Interested in life”	11.200; 0.004 *	−3.060; 0.002 *	1.021; 0.307	−2.283; 0.022 *
“Satisfied”	13.931; <0.001 *	−2.353; 0.019 *	−0.602; 0.547	−3.509; <0.001 *
“That you had something important to contribute to society”	21.529; <0.001 *	−1.200; 0.230	−2.553; 0.011 *	−4.628; <0.001 *
“That you belonged to a community”	15.727; <0.001 *	−2.660; 0.008 *	−0.435; 0.664	−3.652; <0.001 *
“That our society is becoming a better place”	48.670; <0.001 *	−6.540; <0.001 *	2.522; 0.012 *	−4.435; <0.001 *
“That people are basically good”	51.823; <0.001 *	−6.062; <0.001 *	1.121; 0.262	−6.062; <0.001 *
“That the way our society works made sense to you”	59.217; <0.001 *	−6.225; <0.001 *	0.751; 0.453	−6.313; <0.001 *
“That you liked most parts of your personality”	5.167; 0.075	N/A **		
“Good at managing the responsibilities of your daily life”	9.927; 0.007 *	−2.512; 0.012 *	3.090; 0.002 *	0.973; 0.330
“That you had warm and trusting relationships with other kinds”	4.033; 0.133	N/A **		
“That you had experiences that challenged you to grow or become a better person”	7.139; 0.028 *	−2.243; 0.025 *	2.575; 0.010 *	0.637 0.524
“Confident to think and express your own ideas and opinions”	3.787; 0.144	N/A **		
“That your life has a sense of direction or meaning to it”	9.004; 0.011 *	−2.312; 0.021 *	−0.101; 0.920	−2.564; 0.010 *
Overall emotional	16.435; <0.001 *	−2.912; 0.004 *	−0.713; 0.863	−3.620; <0.001 *
Overall social score	59.283; <0.001 *	−5.952; <0.001	0.304; 0.761	−6.566; <0.001 *
Overall psychological score	8.908; 0.012 *	−2.982; 0.003 *	1.894; 0.058	−2.952; 0.003 *
Overall MHC-SF score	31.287; <0.001 *	−4.417; <0.001 *	−0.881; 0.378	−4.391; <0.001 *

* Statistically significant; *p* < 0.05. ** N/A = not applicable; the Dunn’s test was not conducted when the *p*-value of the KW test was non-significant.

**Table 5 ijerph-19-07438-t005:** Results of the FS.

Item	KW; *p*-Value	Pairwise Comparisons Dunn’s Test; *p*-Value		
	S1–S2–S3	S1–S2	S2–S3	S1–S3
“I lead a purposeful and meaningful life”	16.042; <0.001 *	−1.433; 0.152	−1.815; 0.070	−4.004; <0.001 *
“My social relationships are supportive and rewarding”	19.468; <0.001 *	−0.723; 0.470	−2.750; 0.006 *	−4.346; <0.001 *
“I am engaged and interested in my daily activities”	48.076; <0.001 *	−2.376; 0.017 *	−3.241; 0.001 *	−6.933; <0.001 *
“I actively contribute to the happiness and wellbeing of others”	4.710; 0.095	N/A **		
“I am competent and capable in the activities that are important to me”	19.145; <0.001 *	−2.500; 0.012 *	−0.995; 0.320	−4.217; <0.001 *
“I am a good person and live a good life”	9.337; 0.009 *	−2.365; 0.018 *	−0.148; 0.882	−2.610; 0.009 *
“I am optimistic about my future”	7.209; 0.026 *	−1.355; 0.175	−0.824; 0.410	−2.653; 0.008 *
“People respect me”	1.339; 0.512	N/A **		
Overall score	37.963; <0.001 *	−2.001 0.045 *	−2.984; 0.003 *	−6.161; <0.001 *

* Statistically significant; *p* < 0.05. ** N/A = not applicable; the Dunn’s test was not conducted when the *p*-value of the KW test was non-significant.

**Table 6 ijerph-19-07438-t006:** Results of the FOHS.

Item	KW; *p*-Value	Pairwise Comparisons Dunn’s Test; *p*-Value		
	S1–S2–S3	S1–S2	S2–S3	S1–S3
“Something might happen at any time and we could easily lose our happiness”	0.452; 0.798	N/A *		
“Happiness is fragile”	2.695; 0.260	N/A *		
“It is likely that our happiness could be reduced to unhappiness with a simple accident”	0.595; 0.743	N/A *		
“There is only a thin line between happiness and unhappiness”	5.453; 0.065	N/A *		
Overall score	1.529; 0.466	N/A *		

* N/A = not applicable; the Dunn’s test was not conducted when the *p*-value of the KW test was non-significant.

**Table 7 ijerph-19-07438-t007:** Spearman correlations between overall scores.

	S1		S2		S3	
Correlations between:	Correlation	*p*-Value	Correlation	*p*-Value	Correlation	*p*-Value
MHC-SF and FS	0.584	<0.001 *	0.679	<0.001 *	0.753	<0.001 *
MHC-SF and FOHS	−0.347	<0.001 *	−0.411	<0.001 *	−0.281	<0.001 *
FS and FOHS	−0.181	<0.001 *	−0.420	<0.001 *	−0.163	0.006 *

* Statistically significant; *p* < 0.05.

## Data Availability

The data is not made public for confidentiality reasons. Some or all of data that supports the findings of this study are available from the corresponding author upon reasonable request.

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
