# Peer review of "Comparing Mental Health, Wellbeing and Flourishing in Undergraduate Students Pre- and during the COVID-19 Pandemic"

_ijerph, 2022, doi:10.3390/ijerph19127438_

Round 1

Reviewer 1 Report

The manuscript “Comparing mental health, wellbeing and flourishing in undergraduate students pre- and during the COVID-19 pandemic” presents data from South African students to investigate possible differences in self-reported indicators of well-being, flourishing, and fragility of happiness. Comparisons are made on the basis of three different groups of students assessed in 2019 and 2021. Results indicate limited effects of the pandemic on these indicators.

Though I found some of the results presented interesting, I do not think the manuscript is fit for publication in its current condition. Additionally, I believe my concerns with the manuscript require such extensive changes that they can not be addressed sufficiently in a revision and have, therefore, decided to recommend a rejection. I will list my concerns below, starting off with three major concerns and followed by a short list of minor concerns. Each single major concern might be mitigated (though not disbanded) in a revision, but the overall combination of these points, I find, make this manuscript unacceptable.

Major concerns:

1) The manuscript states a single hypothesis: that there will be differences between three groups – depending on the date they were assessed. I find this to be wholly unsatisfactory in two major ways. First, if there are, as the authors state in the abstract,a preponderance of studies on student mental health, wellbeing and flourishing during the COVID-19 pandemic”, one would assume it is possible to derive more well-defined hypotheses. Especially when there are three general constructs under investigation, it would seem possible to at least predict the direction of effects for each of them on the basis of this mountain of literature (as I will point out in my minor concerns, the MHC-SF could additionally be split along the three facets it assesses). Since this was not done prior to investigation of the data, this problem cannot be fixed in a revision (as this would constitute HARKing). Second, if one hypothesis is stated, one hypothesis must be tested in the data analysis. Instead, at least 29 “omnibus” hypotheses are tested using the KW-Test without any indication of adjusting for Type-I-Error inflation (which at this point would have accumulated to approximately 0.77 instead of the stated 0.05 used as a decision criterion). I would suggest using the multivariate KW test (Katz & McSweeny, 1980; He et al., 2018) instead.

2) In Section 3.1 (and more specifically, Table 1) the authors present clear evidence of structural differences between the groups that were assessed at T1 and T2 and the group that was assessed at T3 with respect to demographic variables. These differences must be accounted for, when comparing the groups with respect to the variables assessed. Simple weighting approaches are widely available and applicable to many different approaches to comparisons of central tendency – I do not see any argument or reason provided in the manuscript, why this was not done.

3) The results are not interpreted correctly in the discussion. Instead, the discussion seems to be derived mainly from the narrative of adverse mental health effects of the pandemic which was identified from the literature than from the results presented in this manuscript. The very first sentence of the discussion states: “Results indicate that student mental health and wellbeing have predominantly declined significantly during the COVID-19 pandemic […]” This is simply incorrect. Generally two overlapping effects are investigated in this analysis and they are not disentangled properly by the authors (though they are clearly visible in the results). Differences between T1 and T2 constitute general effects of the first year of university (something the Authors investigated in depth in a previous article) – i.e. differences on the outcomes are interpreted as differences between students in their first semester and students in their second semester. Because these two assessment were done in 2019 this effect is free of any pandemic effect. The students assessed at T3 were all assessed in their second semester, thus any pandemic-effect would be visible only in differences between T2 and T3 in this design. Differences between T1 and T3 are overlapping effects of the first to second semester changes already indicated in T1-T2 differences and pandemic effects already indicated in the T2-T3 effects. Focusing on the best indicator of actual pandemic effects provided in this study (differences between T2 and T3) reveals differences for only 6 of 26 indicators investigated here (again, this is without actually correcting for Type-I-Error inflation). Additionally, as the authors then go on to state, three of those indicators actually show increases – i.e. the exact opposite of the effect asserted at the beginning of the discussion (and in the abstract)! Thus, the results of the study are that on three of the 26 indicators investigated, the stated “predominant decline in student mental health” could be found. Please be aware that this is not simply a wrong statement made once – the same interpretation is presented again for the flourishing scale, where differences between T2 and T3 are significant on 2 of the 8 items and this is stated as a “consistent downward trend”.

Minor concerns:

- In the abstract it is stated, that there is a preponderance of studies on this topic. Quantitative proof of this claim (e.g. the number of articles identified under the given keywords in the popular literature databases) would be more than appreciated.

- Citation style is not in line with journal guidelines (https://www.mdpi.com/journal/ijerph/instructions)

- The findings reported on p. 2 as stemming from Løset et al. (2022) are highly cherry-picked from among a plethora of non-significant findings. These should be discussed as well, because the current introduction may be misleading into a false sense of stability and replicability of these effects across studies (additionally, the paper by Løset et al., 2022, also did not correctly adjust for Type-I-Error inflation).

- The sentence on p. 2, l. 57-60 implies that the results regarding decreases in energy and appetite stem from Tran et al. (2022), which would be results from 1851 participants, but in reality they are from Albright et al. (2022), which stem from a much less robust 81 participants. Please rephrase.

- Overall, the introduction should also include a discussion of the wide variety of study quality and sample sizes found throughout the cited papers.

- p. 3: I believe the T1, T2, T3 denotation is misleading, because it implies some form of temporal relation, when in truth these are not “timepoints” of the same study, but different samples.

- p. 3: Footnote indicators are not typeset correctly in the text.

- p. 4, l. 148: Validity cannot be established via a CFA. Factor analysis does not allow for statements about internal validity, external validity, predictive validity, …

- Later analyses all account for non-normality of the data, yet there is no indication this was also done for the CFA. Especially since the CFA relies on single indicators, analyses for ordinal indicators should be conducted instead of the CFA for continuous, normally distributed variables reported here.

- Why are group comparisons not made within the CFA context, when these analyses were already being performed? Multi-Group CFAs would have allowed for more robust tests of not just mean differences but also construct structure (e.g. different factor loadings) which could have generated many additional and interesting results.

- p. 7, l. 244-246: the sentences ends nowhere – please rephrase.

- Section 3.3: the MHC-SF is deemed to be structured into a “general factor” and three specific content factors in the CFA section. I would suggest performing multivariate KW Tests for each of these content factors (emotional, social, and psychological well-being) to at provide some addtional structure to these results and the discussion, instead of reporting only all single-item comparisons and the comparison of the overall-score, which does not seem to correctly account for the results of the CFA (multiple dimensions, item-specific factor loadings).

- p. 12: the correlations between the constructs should be tested for equality across the three samples to provide an indication of comparability.

- I tackled the discussion in-depth in my major concerns and will therefore not list minor concerns for this section separately.

Reviewer 2 Report

Dear authors. 

You did a good job, and it was an honour to review this paper. However, I have some comments and suggestions that, in my opinion, will improve the manuscript. 

So:

1) There aren´t in any part of the document references to the ethical approval by the Ethics Committee;

2) You should write in the section where you introduce the instruments if they have a cut-off.

3) If the participants are not the same in the three moments, I do not understand why you use the two first moments. What do you want to analyse with this comparison? It must be explained. It would be enough to compare the moment T2 and T3 because the T2 is the closest to the pandemic's beginning. 

4) It would be vital if you could analyse the data with multivariate analysis like Logistic Regression or Linear Regression using or not categorical variables dummies. 

5) In all analyses, you use non-parametric tests, which is good. However, why do you use the mean and standard deviation to describe the scores obtained? If the distributions are not normal, you must use the median and interquartile range.

6) In the study's limitations, you do not include the participants not being the same in the three moments. This is the main limitation facing the goal of the study. 

7) There is no congruence in the norm in the references list. Some have DOI, and some don't. Please verify because the majority of these cases are articles in scientific journals.

Reviewer 3 Report

This is a very well researched study and a well written manuscript. The introduction section is well written except I thought that the authors could separate the mental health and well-being as two separate constructs rather than combining them together.

In the Methods section under ' Data Analysis' lines 183-184 refer to Table 2. I think this needs to be ' Table 4'.

The results and discussion section are written clearly and include good evidence. Line 422 mentions that ' flourishing' should be reestablished by having supportive environments but the authors haven't included how this could be done?  It may be important to make some suggestions for this.

It may be a good idea to add a future research section or add this to the conclusion section of the authors envisage studies to be designed in future which could overcome the limitations of this design and make them robust.
